# Combining Experimental Evolution and Genomics to Understand How Seed Beetles Adapt to a Marginal Host Plant

**DOI:** 10.3390/genes11040400

**Published:** 2020-04-08

**Authors:** Alexandre Rêgo, Samridhi Chaturvedi, Amy Springer, Alexandra M. Lish, Caroline L. Barton, Karen M. Kapheim, Frank J. Messina, Zachariah Gompert

**Affiliations:** 1Department of Biology, Utah State University, Logan, UT 84322, USA; alexandre.rego@aggiemail.usu.edu (A.R.); amy.springer@aggiemail.usu.edu (A.S.); alex.lish@aggiemail.usu.edu (A.M.L.); clbourgeois@hotmail.com (C.L.B.); karen.kapheim@usu.edu (K.M.K.); frank.messina@usu.edu (F.J.M.); 2Department of Zoology, Stockholm University, 114 19 Stockholm, Sweden; 3Department of Organismic & Evolutionary Biology, Harvard University, Cambridge, MA 02138, USA; samridhi.chaturvedi@gmail.com

**Keywords:** plant-insect interaction, host shift, parallel evolution, detoxification, experimental evolution, population genomics, genome-wide association mapping, gene expression, *Callosobruchus maculatus*

## Abstract

Genes that affect adaptive traits have been identified, but our knowledge of the genetic basis of adaptation in a more general sense (across multiple traits) remains limited. We combined population-genomic analyses of evolve-and-resequence experiments, genome-wide association mapping of performance traits, and analyses of gene expression to fill this knowledge gap and shed light on the genomics of adaptation to a marginal host (lentil) by the seed beetle *Callosobruchus maculatus*. Using population-genomic approaches, we detected modest parallelism in allele frequency change across replicate lines during adaptation to lentil. Mapping populations derived from each lentil-adapted line revealed a polygenic basis for two host-specific performance traits (weight and development time), which had low to modest heritabilities. We found less evidence of parallelism in genotype-phenotype associations across these lines than in allele frequency changes during the experiments. Differential gene expression caused by differences in recent evolutionary history exceeded that caused by immediate rearing host. Together, the three genomic datasets suggest that genes affecting traits other than weight and development time are likely to be the main causes of parallel evolution and that detoxification genes (especially cytochrome P450s and beta-glucosidase) could be especially important for colonization of lentil by *C. maculatus*.

## 1. Introduction

Genomic approaches have identified genes responsible for adaptive evolution in natural populations and experimental lines [1]. Examples include genes with large effects on defensive armor in sticklebacks (*Eda* and *Pitx-1*) [2,3], coat color in mice (*Agouti* and *Mc1r*) [4,5], wing pattern in *Heliconius* butterflies (*Optix* and *WntA*) [6,7,8], and diapause in *Drosophila* (the *insulin-regulated PI3 kinase* gene and *Timeless*) [9,10]. Although genomic approaches have been less successful at determining the specific genes affecting highly polygenic traits, these methods still can elucidate trait genetic architecture, such as the number of quantitative-trait loci (QTL) or causal variants, trait heritabilities, and genetic covariances among traits [11,12,13]. Summaries of trait genetic architecture can help explain patterns and dynamics of evolutionary change (e.g., [12,14,15]). Nonetheless, most empirical work on the genetics of adaptation comes from a modest number of model systems, and even in these systems the genetic basis of the full suite of traits selected during a bout of adaptation is rarely analyzed. Consequently, generalizations about the relative importance of standing variation versus new mutation, partial versus complete selective sweeps, the number and effect sizes of genes responsible for adaptation, and the repeatability of adaptive evolution remain preliminary and contentious (e.g., [1,16,17,18,19]).

A major ongoing debate concerns when, whether, and to what extent the same genes or mutations are selected when multiple populations or species adapt to the same or similar environment, i.e., how parallel is evolution at the genetic level [20]. Numerous examples of parallel, adaptive genetic changes have been documented (e.g., [21,22,23,24,25,26]), but parallelism is rarely complete and there are numerous counter-examples of more idiosyncratic or unpredictable evolution (e.g., [27,28,29]). Several factors could limit parallelism at the genetic level, including differences in trait genetics or selective pressures. For example, populations may harbor different standing genetic variation for traits and different specific mutations may arise during adaptation [30,31]. Evidence for non-parallel mechanisms includes finding different QTL for the same traits across populations or environments [32]. Even when the same genes affect traits in multiple populations, the same genetic changes might not be observed if selection varies because of large or small environmental differences. For these reasons, experimental evolution serves as a well-controlled means to assess the degree of parallelism underlying adaptation (this differs from using parallelism as a test for selection, e.g., [33]).

More generally, studies that combine population-genomic analyses with trait genetic mapping provide powerful opportunities to shed light on the genetic basis of adaptation, including potential constraints on parallel evolution [19]. Population-genomic approaches include genome scans for detecting selection and the use of evolve-and-resequence (E&R) experiments (e.g., [34,35,36]). These approaches are similar in that they examine the outcomes of evolution and thus inherently integrate across the various sources and targets of selection [37,38]. Experimental evolution studies typically include replication, precise control of environmental conditions, known demographic conditions, and temporal sampling, all of which aid in identifying genes responsible for adaptation (e.g., [33,37,39,40,41]). This tractability of course comes at the cost of ambiguous relevance to natural populations. Moreover, these population-genomic approaches often do not explicitly connect adaptive genetic changes to specific traits (but see, e.g., [42,43]).

Genotype-phenotype association mapping, either in natural populations or experimental crosses, provides a more direct way to determine the genetic basis of trait variation within or between populations (e.g., [12,44,45]). With such methods, the genotype-trait connection is explicit, albeit for only a subset of possible adaptive traits. Mapped populations might not experience the same environmental conditions as natural populations (or even experimental lines), and thus might not exhibit the same genotype-phenotype associations if these are environment dependent (e.g., [46]). Thus, population-genomic analyses and trait mapping approaches should be viewed as complementary, and combining the two methods is especially likely to reveal the genetic and phenotypic bases of adaptation (e.g., [29,47,48,49,50,51]). Lastly, an even more comprehensive analysis would be to combine gene-expression data and genetic manipulations (e.g., RNA interference or genome editing) with population-genomic or trait-mapping data [3,52,53]. Gene expression data and genetic manipulations add a more mechanistic understanding to the link between genotype and phenotype [54,55,56], and in some cases can identify which genotype-phenotype associations are causal (e.g., [57]).

Here, we combine E&R experiments, genome-wide association (GWA) mapping, and analyses of gene expression to examine the genetic basis of adaptation to a marginal host plant by the seed beetle *Callosobruchus maculatus*. The E&R experiments and GWA mapping consider three distinct selection lines, and thus allow us to assess parallelism in terms of both patterns of evolutionary change and genotype-phenotype associations. The gene-expression data are not similarly replicated, but provide additional functional information to complement the population-genomic and mapping analyses.

The cowpea seed beetle *Callosobruchus maculatus* infests human stores of grain legumes. It has long served as a model system for investigating the evolution of insect-plant interactions [58,59,60], and has been used more recently to examine a variety of questions in evolutionary biology, especially those involving sexual selection and sexual conflict (e.g., [61,62,63,64,65]). Because of current advances in genomic resources, this system is poised as an emerging model system for evolutionary genomics as well [38,66,67].

Female beetles attach eggs to the surface of legume seeds. Hatching larvae burrow into the seed and must complete development in the single, natal seed. Because *C. maculatus* has been associated with stored legumes for thousands of years, laboratory conditions are a good approximation of its “natural” environment [68]. Beetle populations mainly attack grain legumes in the tribe Phaseoleae, particularly those in the genus *Vigna* [69]. Lentil (*Lens culinaris*), a member of the tribe Fabeae, is a very poor host for most *C. maculatus* populations, as larval survival in seeds is typically <5% [38,70,71,72]. Nonetheless, lentil is used as a host by a few unusual ecotypes in certain regions [73,74].

Attempts to establish laboratory populations on lentil have often resulted in extinction [73], but in a minority of cases experimental lines have rapidly adapted to this host [38,71]. In a South Indian *C. maculatus* population (denoted M) that was collected from and maintained on mung bean (*Vigna radiata*), survival on lentil in four experimental lines increased from <5% to >80% within 20 generations, with much of the change occurring in the first five generations [38,71]. This increase in survival was accompanied by increased adult weight and decreased development time on lentil (two other metrics of performance) [71]. Rapid adaptation caused evolutionary rescue; the four lines (previously denoted as L1, L2, L3, and L14) rebounded demographically and have since persisted on lentil, which is now nearly as suitable as the ancestral host, mung bean. Population-genomic analyses of these lines showed a mixture of parallel and idiosyncratic allele frequency changes driven by selection on lentil [38,40]. We found evidence of genetic trade-offs, whereby alleles favored by selection on lentil were selected against on mung bean, and of multiple genetic regions associated with adaptation to lentil [38,40]. Our analyses of a fine-grained, population-genomic time series from one of these lines (L14) documented strong selection and rapid evolution at multiple loci with allele frequency changes of ∼0.4 in a single generation, and with some initially uncommon alleles fixing (or nearly fixing) in as few as five generations [38]. This past work was based on a highly fragmented genome assembly, and did not connect selection in the E&R experiments to trait or functional information. Thus, much remains unknown about the genomic basis of rapid adaptation to lentil.

In the current study, we build on past work by re-analyzing population-genomic data from the E&R experiments in the context of an improved genome assembly and annotation [67]. We combine this reanalysis with genome-wide association mapping of two performance traits (adult weight and development time) and with gene-expression data. We ask the following specific questions: (i) How are genetic regions that evolved rapidly during lentil adaptation distributed across the genome, and how does this vary across replicate lines?, (ii) Do the same loci affect adult weight and development time in different lentil lines, and do these two performance traits share a common genetic basis?, (iii) To what extent do loci associated with weight and development time occur in parts of the genome known to have evolved rapidly during lentil adaptation?, and (iv) To what extent do differences in host environment and evolutionary history (i.e., genetics) result in differences in gene expression, and do differentially expressed genes co-localize with SNPs associated with weight and development time or with genetic regions of rapid evolution during lentil adaptation? By integrating these three sets of genomic data, we are able to quantify parallelism in terms of the genetic basis of performance traits and allele-frequency change during adaptation (we test for parallelism in each of these patterns rather than using parallelism as a test for the process of selection), and obtain a more complete understanding of the genes and traits that allow *C. maculatus* to adapt to and persist on a marginal host plant. In terms of the latter, we specifically test for roles of digestive enzymes and detoxification genes as these types of genes have been implicated in host-plant adaptation in other systems [53,75,76].

## 2. Materials and Methods

We analyzed six experimental lines in the current study: the M line, which was originally collected from South India and has since been maintained in the lab on its ancestral mung-bean host [77,78], three lentil-adapted lines (L1, L2, and L14, each independently derived from M), and two reversion lines (L1R and L2R) that were switched back to mung bean after many generations on lentil (Figure 1, Appendix A). The South India M line has been maintained at a census population size of 2000–2500 individuals for >300 generations; past genetic analyses suggest a variance effective population size of ∼1149 [40]. Details on the establishment of L1, L2 and L14 can be found in [40,71] (L1 and L2) and [38] (L14). The reversion lines, L1R and L2R, were initiated to test for genetic trade-offs between performance on mung bean versus lentil. These lines were shifted back onto the ancestral host in order to examine whether there would be a decrease in the ability to use lentil (as predicted by a trade-off hypothesis) [79]. Thus, allele frequency change in the lentil lines should reflect adaptation to lentil (and genetic drift), whereas changes in the reversion lines relative to their source lentil lines should reflect adaptation back to mung bean (and perhaps drift to a lesser extent) (past work has attempted to parse the roles of selection and drift [38,40], but here we simply focus on change). Herein, we analyze patterns of genome-wide allele frequency change for combinations of all six of these lines (we ignore two additional lines, L3 and L3R, as we lack trait-mapping data for these lines). Trait-mapping data come from backcross mapping populations created by crossing M with L1, L2 and L14 (denoted BC-L1, BC-L2, and BC-L14). Gene expression data come from M, L1 and L1R, that is from the source mung bean line, a lentil line, and its corresponding reversion line. We measured gene expression in all three lines when reared in mung bean (L1${}^{M}$, L1R${}^{M}$, M${}^{M}$), and for L1 and L1R when reared in lentil (L1${}^{L}$, L1R${}^{L}$) (rearing the M line on lentil for expression data was not possible given the extremely low survival rates).

### 2.1. Evolve-and-Resequence Experiments

Each lentil-adapted line was established using the same protocol described by [71]. Briefly, a line was formed by adding >2000 (L1) or >4000 (L2, L14) adults to 1500 g of lentil seeds. All lines experienced a severe initial bottleneck, with an initial survival within seeds of 1–2% [38,71]. Whereas most attempts to establish beetle populations on lentil failed, survival rates increased rapidly in these three lines. After census population sizes recovered from the initial bottleneck, each successive generation for each line was formed by adding >2000 adults to 750 g of lentil seeds [38,71]. Lines L1 and L2 were formed in 2004, but beetles were not used for DNA sequencing until much later (after >70 generations had elapsed). In contrast, the L14 line was formed in 2014, and we sampled and sequenced beetles during each generation throughout the early stages of lentil adaptation. L14 was split into two sublines (L14A and L14B) after the F4 generation. The two sublines exhibited highly parallel evolutionary changes in allele frequencies, and here we focus primarily on L14A [38].

Reversion lines, in which lentil-adapted lines were reverted back to mung bean, were established once the lentil-adapted populations had reached a plateau in fitness on lentil (as measured by survival during performance assays [71,79]). Lentil lines were reverted to mung bean at F62 for L1, and at F48 for L2 (Figure 1, Appendix A). The difference in time of reversion was simply due to L1 having been established earlier than L2. Each reversion line was generated by transferring >2000 adults to 750 g of mung beans, and the same protocol was used for each successive generation. In this manner, successive generations of both lentil (non-reverted) and reverted lines were formed in the same fashion.

We obtained partial genome sequences from line samples (992 beetles from 22 line × generation combinations, where lines denote the different lentil or reversion lines each founded as described above) using our standard genotyping-by-sequencing approach [80]. With this approach, sequence data can be associated with individual beetles. This includes a total of 1.3 billion, 100 bp single-end DNA sequences. These archived data were used in the current manuscript (NCBI SRA PRJNA480050).

### 2.2. Trait Mapping Experiment

Lines L1, L2, and L14A were used in the creation of the backcross (BC) mapping populations. To generate each population, 12 newly-emerged, unmated females from each lentil line were isolated and each was paired with a newly-emerged adult male from the M line (i.e., each of the 12 pairs comprised an unmated female and a newly-emerged male). We obtained unmated lentil-line females by isolating a few hundred lentil seeds from each stock culture in 4-mL vials. Hybrids were formed when the L1, L2, and L14A lines had spent 146, 135, and 38 generations on lentil, respectively.

Individual pairs of unmated females from the L lines and males from the M line were placed in 60-mm Petri dishes with a single layer of mung beans (about 100 seeds). Dishes were kept in a growth chamber at standard conditions of 25 °C and constant light. Pairs were allowed to mate and lay eggs for 48 h. After 10 days, we inspected the dishes and collected several seeds bearing a single hatched egg (one hybrid larva within the seed) per dish. Seeds were then isolated in 4-mL vials for each cross type.

Newly-emerged, unmated F1 hybrid females were then backcrossed to L1, L2 or L14A males. Isolated pairs (hybrid females and lentil-line males) were allowed to mate and oviposit in dishes containing approximately 100 lentil seeds for 48 h. After 10 days, several hundred lentil seeds bearing a backcross larva (indicated by a single hatched egg on the seed surface) were isolated in 4-mL vials for each cross type. Once backcross adults were about to emerge, we checked vials for emerging adults daily. Emerged adults were sexed and weighed on a Mettler Toledo XPE105 analytical microbalance (Mettler Toledo) to the nearest 0.01 mg. Development time was measured as the number of days between the removal of each parental pair and the emergence of the backcross progeny. Vials were checked until two weeks after the last adult emerged to ensure that the backcross generation had finished emerging. We collected a total of 476, 760 and 529 BC adults from backcrosses to L1, L2, and L14A, respectively (weight and development time data are available from Dryad, https://doi.org/10.5061/dryad.3j9kd51dw). In the subsequent trait-mapping analysis, we only used females to avoid sex effects and hemizygosity for sex chromosomes.

Beetles were then stored at −80 °C prior to DNA extraction. We isolated DNA from 748 female BC beetles: 241 from BC-L1 (mean per family = 20.08, s.d. = 15.81), 256 from BC-L2 (mean per family = 23.27, s.d. = 20.70), and 251 from BC-L14 (mean per family = 20.2, s.d. = 15.78). We used Qiagen’s DNeasy 96 Blood & Tissue Kit (Qiagien Inc., Hilden, Germany). DNA fragment libraries for genotyping-by-sequencing (GBS) from each beetle were prepared using methods described in [80]. Briefly, we used the endonucleases *EcoRI* and *MseI* to digest beetle genomes. We ligated adaptor oligos with internal 8–10 bp barcode sequences and Illumina primer sites using T4 DNA ligase. PCR was used to amplify the restriction fragment libraries. Pooled, barcoded libraries (two pools with half of the beetles in each pool) were purified and size-selected using a BluePippin (Sage Science, Beverly, MA, USA) to retain 250–350 bp fragments. Whereas DNA libraries were pooled, sequences from each beetle could be uniquely identified from individual, internal barcode sequences. These libraries were sequenced over two full runs on an Illumina NextSeq with 150 bp single-end reads. DNA sequencing was performed by the Genomics core lab at Utah State University (Logan, UT, USA). These DNA sequence data have been archived on NCBI’s SRA (PRJNA616195).

### 2.3. Gene Expression Experiment

We established five treatments, L1${}^{L}$, L1${}^{M}$, L1R${}^{L}$, L1R${}^{M}$, and M${}^{M}$, for generating gene-expression data, where each superscript denotes the larval rearing host, lentil or mung bean. At the time of sampling for RNA analysis, the L1${}^{L}$ line has spent 107 generations on lentil, and the L1${}^{M}$ line was switched to mung bean for one additional generation. The reverted L1R${}^{M}$ line had spent 55 generations on mung bean, and the L1R${}^{L}$ line was switched to lentil for one additional generation. The M${}^{M}$ line had been continuously reared on mung bean for >300 generations.

To obtain actively feeding larvae, we closely monitored the appropriate stock culture to determine when the population consisted of mainly 4th-instar (last instar) larvae within seeds. Determining the developmental stages of larvae was accomplished by gently cracking seeds with a small hammer to separate the cotyledons and expose the open larval burrows at the center of the seed. We targeted 4th-instar larvae because that stage is characterized by highly active feeding, just before the prepupal stage. When larvae in a given culture were mostly in the appropriate stage, we again used a small hammer to open seeds. We immediately placed each exposed, feeding larva into a labeled, perforated, 1.5 mL Eppendorf tube, and immersed the tube in thermos filled with liquid nitrogen. After a batch of tubes had been submerged, we poured the liquid nitrogen into a strainer submerged in dry ice, and immediately transferred each tube to a freezer at −80 °C.

We extracted total RNA from five biological replicates of three pooled larvae from each of the five lines (L1${}^{L}$, L1${}^{M}$, L1R${}^{L}$, L1R${}^{M}$, and M${}^{M}$) using RNeasy Mini kits (Qiagen, Inc.) according to the manufacturer’s protocol. RNA was eluted in 50 μL of water and stored at −80 °C. Quality and quantity was evaluated on a TapeStation System (Agilent, Inc.) at the Utah State University Center for Integrated Biosystems.

RNA libraries were prepared with the TruSeq Stranded mRNA Library Construction kit (Illumina, CA, USA) at the Roy J. Carver Biotechnology Center at the University of Illinois at Urbana-Champaign. RNAseq libraries were quantitated by qPCR and sequenced on two lanes of a HiSeq2500 for 101 cycles from each end of the fragments (100 nt paired-end reads) using a HiSeq SBS kit version 4. Fastq files were generate and demultiplexed with the bcl2fastq v1.8.4 Conversion Software (Illumina, CA, USA). We generated 880,655,484 reads, with a mean ± s.d. of 35,226,219 ± 3,061,575 reads per library. These RNA sequence data have been archived on NCBI’s SRA (PRJNA616195).

### 2.4. DNA Sequence Alignment and Variant Calling

We first filtered the backcross lines’ Fastq files to remove PhiX sequences and trim poly-G tails, which arise from missing signal with the 2-dye chemistry used for NextSeq sequencing. We then demultiplexed the Fastq files using custom perl scripts (these scripts are available from Dryad, https://doi.org/10.5061/dryad.3j9kd51dw). We used the mem algorithm from bwa (version 0.7.17-r1188) [81] to align the 936 million, 150 bp DNA sequences from the backcross lines to a recently published, *C. maculatus* genome assembly [67]. We also aligned the Fastq files from our older beetle datasets (L1, L1R, L2, L2R, L14A, L14B, M) to the recent assembly. Default parameters were used for the mem algorithm with the exception of minimum seed length (-k 20) and re-seed threshold (-r 1.3). The new *C. maculatus* genome assembly is a substantial improvement over previous resources for this species (e.g., [38]), with a total size of 1.01 gigabases and N50 of 149 kilobases [67]. Additionally, BUSCO estimates of completeness from sets of preserved proteins are high (75% complete, 10% partially complete). Nonetheless, because of the highly repetitive nature of this genome (>63% repeat content), the genome remains fragmented with many small, sub-chromosomal scaffolds (15,778 scaffolds total).

We ran two sets of variant calling on the alignments, one with all of the samples from the E&R experiments and the BC mapping populations and one without the mapping populations. We used the former for genome-wide association mapping, and the latter for population genomic analyses of the E&R experiments. We used the Bayesian multiallelic/rare variant caller option implemented in samtools (version 1.5) and bcftools (version 1.6). The -C 50 command was used, as recommended for Illumina HiSeq data. Bases with a quality score <30 and reads with a mapping quality <20 were ignored. The prior for θ was set to 0.001 and we only called SNPs when the posterior probability that a nucleotide was invariant was <0.01 (compared to the default, much less stringent option of 0.5).

We then filtered each SNP set to only retain SNPs with minimum coverage ≈2X (per beetle), a minimum number of 10 reads supporting the non-reference allele, a minimum mapping quality of 30, no more than 25% of individuals with missing data, and minimum minor allele frequency of ∼0.005. We chose a minimum of 2X coverage to be consistent with past work, and because even with 2X coverage allele frequencies can be estimated very accurately, especially when using models that account for finite coverage and sequence error when inferring genotypes and allele frequencies as we do (see below) [82]. The cut-off for missing data was chosen to minimize locus drop-out and drop-in, and we allowed for a low minor allele frequency to capture rare variants that can be informative for GWA mapping (e.g., [83]). A second round of filtering was done that filtered by maximum coverage (mean coverage + 2 s.d.) to avoid possible paralogs. We retained 17,098 and 20,376 SNPs after filtering for the set that contained all lines and the set that excluded the BC samples, respectively.

### 2.5. Measuring Evolutionary Change during the Evolve-and-Resequence Experiments

We first quantified the extent of evolutionary change (change in allele frequencies) in the evolve-and-resequence (E&R) experiments. We estimated allele frequencies for each line and sample using a hierarchical Bayesian model (as in [84]). This approach accounts for uncertainty in genotypes (as captured by the genotype likelihoods output from bcftools) when estimating population allele frequencies. We specifically obtained allele frequency estimates with the program popmod (version 0.1, Dryad doi:10.5061/dryad.7b5m7; [85]) using a Markov chain Monte Carlo (MCMC) algorithm with a burn-in of 1000 iterations, followed by 10,000 sampling iterations with a thinning interval of 5.

We then focused on evolutionary change between five specific pairs of lines and samples: (i) M to L1 F100 (change during lentil adaptation for L1), (ii) M to L2 F87 (change during lentil adaptation for L2), (iii) L1 F91 to L1R F46 (change in the L1 reversion line relative to L1), (iv) L2 F78 to L2R F45 (change in the L2 reversion line relative to L2), and (v) L14 P to L14A F16 (change during lentil adaptation for L14). We chose these specific samples as in each case the first sample (hereafter ‘ancestral sample’) is our best approximation of the ancestor of the second, and the second population (hereafter ‘derived sample’) represents our endpoint (given available data) for each selection line. In some cases the first population is the actual ancestor (L14 P), but in most it is a descendant of the ancestor (in these cases the ancestral line had been on the same host, mung bean or lentil, for a sufficient amount of time to be well adapted to that host). The parallel evolution hypothesis predicts correlated patterns of allele frequency change across comparisons (i), (ii) and (v) (i.e., adaptation to lentil), and across comparisons (iii) and (iv) (i.e., reversion on mung bean).

We do not attempt to estimate selection coefficients from these data (this was the focus of past work, e.g., [38,40]), but rather to identify contiguous sets of SNPs exhibiting exceptionally high amounts (or rates) of evolutionary change (see for example [86]), which we then compare with genetic regions associated with performance traits or differential gene expression between hosts and lines (see below). Note, however, that past tests of selection based on parameterized models of genetic drift showed that natural selection contributed to evolutionary change in the lentil and reversions lines, especially at loci exhibiting the greatest allele frequency changes [38,40]. To this end, we first estimated standardized allele frequency change for each locus and pair of samples as $\Delta p_{i}=\frac{p_{i}^{1}-p_{i}^{0}}{\sqrt{2*p_{i}^{0}*\left(1-p_{i}^{0}\right)}}$, where $p_{i}^{1}$ and $p_{i}^{0}$ denote the Bayesian estimates of the allele frequencies in the derived (second) and ancestral (first) sample for each pair of lines and samples given above (the superscripts here are indexes not exponents). The denominator is proportional to the ancestral population expected heterozygosity and the expected variance in allele frequency, and thus to the expected change by drift or selection (conditional on $N_{e}$ and the strength of selection). Thus, this standardizes observed change by the genetic variation at a locus and aids in comparisons across loci. However, because the lines and samples differ in $N_{e}$ and in the time elapsed [38,40], values are not directly comparable across the different line and sample pairs.

We fit hidden Markov models (HMMs) to estimates of standardized allele frequency change to identify sets or runs of linked SNPs exhibiting exceptional rates of change for a given line and sample pair. We defined two hidden states, average change and exceptional change, and assumed that the estimates of standardized allele frequency change followed a normal distribution with a mean and standard deviation dictated by the hidden states. We set the means of the hidden states to the median (average change state) and 99.5th percentile (exceptional change state) of the change estimates for each pair of lines and samples. The empirical standard deviation was used as the standard deviation for each state. We used the R (version 3.6.2) package HiddenMarkov (version 1.8.11) [87] to fit the models, but modified the Mstep function to allow for these fixed parameter values (script available from Dryad, https://doi.org/10.5061/dryad.3j9kd51dw). Doing so allowed us to focus on hidden states of interest for detecting exceptional change. We used the Baum-Welch algorithm with 500 iterations and a tolerance of $1e^{-4}$ to estimate the transition matrix between hidden states and the Viterbi algorithm to estimate the hidden states (average change versus exceptional change). This was repeated twice for each comparison to ensure consistency of the results.

### 2.6. Multilocus Genome-Wide Association Mapping

We first estimated the among family variance in adult weight and development time for each BC mapping population by fitting linear mixed-effect models using a restricted maximum likelihood approach (REML). In our models, no fixed effects were included except for the grand mean (only females were analyzed, so sex was not included in the model). This was done with the lmer function in the lme4
R package (package version 1.1.19, R version 3.4.4; [88]). We tested whether among-family variance significantly deviated from a null expectation of 0 using an exact restricted likelihood ratio test based on 10,000 simulated values ([89,90]. This was done with the exactRLRT function in the RLRsim package in R (version 3.1.3; [91]).

We then fit Bayesian sparse linear mixed models (BSLMMs; [92]) with gemma (version 0.98) to estimate the genetic contribution to variation in adult weight and development time in the BC mapping populations, and to identify specific SNPs associated with variation in these performance traits (again, only female beetles were sequenced and analyzed, so sex was not included in the models). Whereas traditional genome-wide association (GWA) mapping methods test one SNP at a time, this polygenic GWA method fits a single model with all SNPs simultaneously and thus mostly avoid issues related to testing large numbers of null hypotheses. Moreover, unlike standard QTL mapping approaches, this method readily handles mapping populations comprising multiple, heterogeneous families, with functional genetic variants potentially segregating within and among families and parental lines. This feature is highly desirable given our experimental design.

Trait values (weight and development time) are modeled as a function of a polygenic term and a vector of the (possible) measurable effects of each SNP on the trait (β) [92]). A Markov chain Monte Carlo (MCMC) algorithm with variable selection is used to infer the posterior inclusion probability (PIP) for each SNP, that is, the probability that each SNP has a non-zero effect, and the effect conditional on it being non-zero [93]. The polygenic term defines each individual’s expected deviation from the grand phenotypic mean based on all of the SNPs. This term accounts for phenotypic covariances among individuals caused by their relatedness or overall genetic similarity (i.e., observed kinship) [92]. The kinship matrix also serves to control for population structure and relatedness when estimating the effects of individual SNPs (β) along with their PIPs. Similarly, SNPs in linkage disequilibrium (LD) with the same causal variant effectively account for each other, such that only one or the other is needed in the model, and this is captured by the PIPs.

The hierarchical structure of the model provides a way to estimate additional parameters that describe aspects of a trait’s genetic architecture [12,92,93]. These include the proportion of the phenotypic variance explained (PVE) by additive genetic effects (this includes β and the polygenic term, and should approach the narrow-sense heritability), the proportion of the PVE due to SNPs with measurable effects or associations (this is called PGE and is based only on β), and the number of SNPs with measurable associations (n-γ). All of these metrics integrate (via MCMC) over uncertainty in the effects of individual SNPs, including whether these are non-zero. Using this BSLMM approach, it is also possible to obtain genomic-estimated breeding values (GEBVs), that is, the expected trait value for an individual from the additive effects of their genes as captured by both β and the polygenic term [12,13].

We fit BSLMMs for adult weight and development time using gemma (version 0.98; [92]). Trait values for each BC line were normal-quantile transformed prior to analysis. Genotypes were inferred using the admixture model in entropy (version 1.2) [80]. This works with the genotype likelihoods from bcftools and applies locus specific prior information on allele frequencies while accounting for uncertainty in the source population (ancestry) of each allele. entropy was run on the full SNP dataset for each BC line and reference source populations (assuming *k* = 2) source populations (M and L1 F100, L2 F87 or L14A F16 for BC-L1, BC-L2 and BC-L14, respectively). We ran three MCMC chains each comprising a 5000 iteration burn-in, and 10,000 sampling steps with a thinning interval of five. Point estimates of genotypes were then obtained as $\bar{g_{ij}}=\sum_{x}\mathrm{Pr}\left(g_{ij}=x\right)x$, where $g_{ij}$ is the count (0, 1, or 2) of the non-reference allele, and $\mathrm{Pr}(g_{ij}=x)$ is the posterior probability that the genotype = *x*. We fit the BSLMMs based on these estimated genotypes and the transformed trait values with 10 MCMC chains (for each trait and line), each with a 1 million iteration burn-in followed by 1 million sampling iterations with a thinning interval of 20. SNPs with minor allele frequencies less than 0.001 in a given mapping population were dropped from the analysis.

### 2.7. Gene Expression

We used RCorrector [94] to detect unfixable k-mers in the RNA sequences and correct these k-mer based read errors. RCorrector compares k-mer based error correction tools, and identifies whether the read has been corrected or has been detected as containing an uncorrectable error. We then used a custom python script to discard unfixable reads identified by RCorrector. Reads were then trimmed using TrimGalore! (version 0.3.3) [95] to remove Illumina adapter sequences. Filtered and quality-checked paired-end reads were aligned to an existing, annotated transcriptome of *C. maculatus* using STAR (version 1.5.2) [96]. STAR alignment rate ranged between 62–77% for all sample libraries. We converted STAR alignments to gene count data for each sample using featureCounts (version v2.0.0) [97].

We performed filtering and normalization of count data using the edgeR [98] library in R version 3.4.2. We removed genes with low expression levels using edgeR. Specifically, we first normalized our data by calculating counts per million values for each gene (using the cpm function in edgeR) and then retained genes with counts per million values >0.5 in at least two samples. We thus retained 10,802 genes for downstream analyses. We visualized expression variation based on these counts per millions with non-metric multidimensional scaling (NMDS) using the plotMDS function and the heatmap function in R.

We conducted differential expression analyses using the limma library [99] in R. We first performed variance stabilizing normalization of the data using the voom [100] library in R and used a design matrix corresponding to the specific linear model used for the analysis. We used the model.matrix function to fit model design matrices using the host and line as factor combinations and then extracted the comparisons of interests as contrasts using the function makeContrasts. We calculated the significance of model effects using voom precision weights and the eBayes function. We then used the decideTests function to decide if a model effect was significant and retained effects if their Benjamini–Hochberg false-discovery rate corrected *p*-value was less than 0.05. We made contrasts in voom for the following nine pairs of comparisons: (i) L1${}^{L}$ versus L1R${}^{L}$, (ii) L1${}^{L}$ versus L1${}^{M}$, (iii) L1${}^{L}$ versus L1R${}^{M}$, (iv) L1${}^{L}$ versus M${}^{M}$, (v) M${}^{M}$ versus L1${}^{M}$, (vi) M${}^{M}$ versus L1R${}^{L}$, (vii) M${}^{M}$ versus L1R${}^{M}$, (viii) L1R${}^{L}$ versus L1R${}^{M}$, and (ix) L1${}^{M}$ versus L1R${}^{M}$. We used these contrasts to identify the genes which show significant difference in gene expression for each pair of comparison. We then asked whether each of three specific classes of genes were over-represented among differentially expressed genes assuming binomial sampling: cytochrome P450s (known to be involved in detoxification of plant secondary chemicals [53,101]) and two classes of putative digestive enzymes, proteases and carboxylases. We classified genes as likely digestive proteases and carboxylases following the annotated genome provided by [102] (proteases = serine protease, trypsin, chymotrypsin, cathepsin, aspartic proteinase, lysosomal aspartic protease, cysteine protease or proteinase [88 genes]; carboxylases = amylase, cellulase, glucosidase or maltase [29 genes]).

### 2.8. Comparisons across Data Sets

We next turned to comparisons of genomic signals of host use across the three datasets, that is, change from the E&R experiments, genotype-phenotype (weight and development time) associations, and patterns of differential gene expression. We first asked whether and to what extent the QTL density (estimated number of QTL per SNP) for performance traits was greater in SNPs showing exceptionally high allele frequency change than in other parts of the genome. We specifically compared weight and development time mapping results for BC-L1 to change between (i) M and L1 F100 and (ii) L1 F91 and L1R F46, BC-L2 mapping to change between (iii) M and L2 F87 and (iv) L2 F78 and L2R F45, and BC-L14 mapping to change between (v) L14 P and L14A F16. Thus, each comparison was between the mapping results from a given backcross line and the E&R experiment involving the same lentil line. In each case, we calculated the density of QTL across exceptional-change SNPs (based on the HMM hidden states) by calculating the mean PIP for weight or development time over these SNPs. We then obtained null expectations by randomizing the location of the exceptional-change SNPs in the genome and repeating the QTL density calculation. Randomizations involved shifting the SNP coordinates to retain patterns of autocorrelation along genome scaffolds in the original data. We conducted 1000 randomization for each of the five comparisons enumerated above. Calculations and randomizations were conducted in R (version 3.6.2).

We then asked whether differentially expressed genes contained an excess of exceptional-change SNPs (from the HMM and E&R experiments) or an elevated density of performance-trait QTL. We focused on the five gene expression comparisons most relevant for host adaptation: (i) L1${}^{M}$ versus L1${}^{L}$ (plasticity in L1), (ii) L1R${}^{M}$ versus L1R${}^{L}$ (plasticity in L1R), (iii) L1${}^{L}$ versus L1R${}^{L}$ (evolved, genetic differences for expression on lentil), (iv) M${}^{M}$ versus L1${}^{M}$ (evolved, genetic differences for expression on mung), and (v) L1${}^{M}$ versus L1R${}^{M}$ (evolved, genetic differences in expression on mung). For comparisons with allele frequency change in the E&R experiments, we considered change from M to L1 F100 and from L1 F91 to L1R F46 (i.e., the same lines used for gene expression). We determined the number of exceptional-change SNPs within differentially expressed genes, and computed null expectations for overlap using a randomization test (1000 randomization). Randomizations were conducted by shifting the HMM states across SNPs to retain autocorrelation in state. This procedure was repeated for each of the five expression comparisons each of the two evolutionary change comparisons. A similar procedure was used to test for higher QTL density in differentially expressed genes. We focused on the same gene expression comparisons, and considered SNP PIPs for adult weight and development time in BC-L1 (as the expression data were from L1 and L1R). We computed the density of QTL from the SNP PIPs in differentially expressed genes, and compared this to null expectations from 1000 randomizations of the PIPs as described above for the E&R results.

## 3. Results

### 3.1. Evolutionary Change

Lines differed in the extent and variability of allele frequency change during the E&R experiments as expected given the differences in $N_{e}$ and generations elapsed [38,40] (Figure 1). Average allele frequency changes were 0.351 (M to L1 F100, s.d. = 0.465), 0.287 (M to L2 F87, s.d. = 0.377), 0.210 (L1 F91 to L1R F46, s.d. = 0.357), 0.186 (L2 F78 to L2R F45, s.d. = 0.253), and 0.313 (L14 P to L14A F16, s.d. = 0.409). Parallel (similar) patterns of change, as captured by correlations in standardized allele frequency changes, were observed in some cases (Appendix A). Parallelism was most evident in patterns of change for M to L1 F100 and L14 P to L14A F16 (Pearson *r* = 0.46, 95% CI = 0.45–0.47), whereas correlations in standardized change were lower for comparisons between lines adapting to lentil and reversion lines evolving on mung bean.

Based on the HMM fit of standardized allele frequency changes ($\Delta p_{i}$), less than 1% of SNPs were assigned to the exceptional change state in each pair of lines and samples (minimum = 0.5% in M to L2 F87, maximum = 0.9% in L1 F91 to L1R F46) (Figure 2). Exceptional-change SNPs were widely dispersed across the genome with 87 (M to L1), 83 (M to L2), 118 (L1 to L1R), 97 (L2 to L2R) and 95 (L14) contiguous regions comprising 1.3, 1.3, 1.5, 1.5, and 1.3 SNPs on average, respectively. The small size of these HMM regions is consistent with our high estimates of transition rates between hidden states (Appendix A). Akin to patterns of parallelism described above, the greatest correlation in HMM states was for change in M to L1 F100 versus L14 P to L14A F16 (Pearson *r* = 0.27, 95% CI = 0.26–0.28) (Appendix A). We detected an excess of exceptional-change SNPs on the X sex chromosome in M to L2 F87 (randomization test, obs. = 16, expected = 8.9, *P* = 0.004) and L14 P to L14A F16 (obs. = 20, expected = 9.7, *P* = 0.001), but not in the other comparisons. In fact, in the L2 F78 to L2R F45 reversion comparison, we detected a deficit of exceptional-change SNPs on the X (obs. = 4, expected = 12.2, *P* = 0.005).

### 3.2. Multilocus Genome-Wide Association Mapping

Female beetles from the BC-L1 and BC-L14 mapping populations were larger but developed more slowly than those from BC-L2. Mean weight at emergence for BC-L2 was 3.95 mg (s.d. = 0.568) versus 4.13 mg (s.d. = 0.726) and 4.18 mg (s.d. = 0.653) for BC-L1 and BC-L14. Mean development time for BC-L2 was 24.36 d (s.d. = 3.123) versus 25.46 d (s.d. = 2.981) and 25.49 d (s.d. =3.061) for BC-L1 and BC-L14. For each of the three mapping populations, between 7.4 (BC-L1 and BC-L2) and 9.7% (BC-L14) of the variation in adult weight was partitioned among mapping families (Appendix A). However, only BC-L1 exhibited a non-trivial amount of among-family variation in development time (6.6%).

Genetic variation in the backcross mapping families explained 14–38% of the phenotypic variation in female weight, but only 8–9% of the variation in development time (Figure 3, Appendix A). We detected the greatest contribution of genetic variation to adult weight in BC-L1 (PVE = 38%, 95% equal-tail probability interval [ETPI] = 16–61%), whereas phenotypic variation in weight in the other two lines and development time in all lines had smaller genetic components. We failed to clearly parse the relative contributions of genetic loci with infinitesimal versus measurable effects (see the large 95% ETPIs on PGE in Appendix A).

Point estimates of the number of causal variants or QTL ranged from 11 to 35, but also exhibited substantial uncertainty (Appendix A). We found little evidence of SNPs/QTL with highly credible, large effects on either weight or development time, with a slight exception of weight in BC-L2 (Appendix A). Consistent with this, there was a strong relationship between the estimated number of causal variants or QTL on each genome scaffold and the size of the genome scaffold, with the latter explaining >95% of the variation in the former in all cases except BC-L2 (77.7%) (Figure 4). Such patterns are expected for highly polygenic traits that lack major effect loci [12]. The number of QTL within genes and on the X sex chromosome (based on the sum of PIPs for these regions) were consistent with null expectations from randomization tests (all P> 0.1) (see Appendix A for genes overlapping with exceptional-change SNPs in multiple lines).

Genetic correlations based on GEBVs were negative for adult weight versus development time in all lines: r=−0.568 for BC-L1 (95% confidence interval [CI] =−0.650, −0.473), r=−0.652 for BC-L2 (95% CI =−0.718, −0.574), and r=−0.764 for L14 (95% CI =−0.811, −0.707). Correlations in estimated effects, both between traits within lines and for either trait across lines, were generally much lower (Appendix A). However, in some cases, breeding values estimated from genotype-phenotype associations in different mapping populations were correlated to a non-trivial extent (Appendix A). Some of these correlations were positive, but others were negative, which might be expected if the same causal variants existed in multiple mapping populations and were in LD with the same SNP loci, but not necessarily associated with the same SNP alleles.

### 3.3. Gene Expression

Between 10 and 15 million reads were uniquely mapped by STAR for each of the 25 samples. Following filtering of genes in edgeR based on counts per million values, we retained 10,802 genes for downstream analyses. Samples from some line × host combinations clustered together based on NMDS, most notably the M line on mung bean (M${}^{M}$) (Figure 5A). L1R on mung also formed a tight cluster, whereas expression for samples reared on lentil (L1${}^{L}$ and L1R${}^{L}$) was more variable.

A subset of genes were differentially expressed between each host and line combination (Figure 5B, Appendix A). We detected the greatest number of differentially expressed genes between the M line and L1 or L1R (>1000, with the exception of M${}^{M}$ versus L1${}^{M}$), with considerably fewer differentially expressed genes between host treatment for either L1 or L1R (Figure 5B). Thus, differential expression mostly resulted from evolved differences between lines rather than from plasticity caused by the host environment. However, cytochrome P450s (which commonly function in detoxification) were over-represented among differentially expressed genes between hosts in both the L1 and L1R lines (Table 1). This was not true for contrasts between lines on the same host. We saw only suggestive evidence of proteases being over-represented among differentially expressed genes, specifically in comparisons of M to L1M (*P* = 0.07), L1R${}^{M}$ (*P* = 0.11) and L1R${}^{L}$ (*P* = 0.11) (Appendix A). This signal mostly involved cathepsin genes, and to a lesser extent chymotrypsin and serine proteases. Likewise, there was suggestive evidence of excess differential expression of carboxylases between M versus L1R${}^{L}$ (*P* = 0.06). This was driven almost entirely by glucosidases (*P* = 0.033 for glucosidases as a single category), especially β-glucosidase (see the Discussion).

### 3.4. Comparisons Across Data Sets

We found an excess of adult weight and development time QTL from the BC-L1 mapping population among the exceptional-change SNPs in the M to L1 F100 E&R experiment (weight, mean PIP per SNP = 0.0054, randomization *P* = 0.002; dev. time, mean PIP per SNP = 0.0038, randomization *P* = 0.006) (Figure 6). However, the same did not hold for the L1 reversion line or for any of the other E&R lines. Similarly, we found an excess of exceptional-change SNPs for M to L1 F100 among the set of differentially expressed genes between L1 on mung bean versus lentil (i.e., L1${}^{M}$ versus L1${}^{L}$, obs. = 3 SNPs, null = 0.24 SNPs, *P* = 0.017), but not for other comparisons (Appendix A). All three exceptional-change SNPs from the HMM fit occurred within the same gene, 5-oxoprolinase (such autocorrelation is accounted for in our randomization test), which was previously shown to be affected by the presence of a secondary metabolite in the diet of *C. maculatus* and is known to be involved in metabolism [103]. Randomization tests showed that the density of adult weight and development time QTL for BC-L1 within differentially expressed genes did not exceed null expectations (Appendix A).

## 4. Discussion

Predicting phenotypes from genotypes has been a central, albeit elusive aim in genomics and evolutionary biology [17,104,105]. Obtaining a comprehensive understanding of the genetic basis of adaptation is even more difficult, as it requires considering many traits and moving from a genotype-phenotype map to a genotype-phenotype-fitness map. Because host shifts in the lab are a good approximation of host shifts by *C. maculatus* in nature and host is a key component of the environment, host adaptation in *C. maculatus* has the potential to serve as a relatively tractable system for uncovering the genetic basis of adaptation. In the current study, we combined population genomic analyses of E&R experiments, genotype-phenotype association mapping, and analyses of differential gene expression to make some progress towards this goal by investigating the genetic basis of adaptation to a marginal host plant. We found mostly different genetic loci associated with adult weight and development time in different lines, and only in some cases were such loci over-represented among genomic regions that evolved rapidly during adaptation to lentil. In fact, we detected more parallelism (i.e., repeatability) across lines (on average) in patterns of allele frequency change during lentil adaptation than in the genetic architecture of these two traits. Likewise, differential gene expression was mostly (but not entirely) unrelated to population genomic patterns or genotype-phenotype associations. Nonetheless, these combined datasets identified several candidate genes or classes of genes likely affecting the ability of *C. maculatus* to use lentil (e.g., detoxification genes). We discuss and interpret these findings below, with an emphasis on constraints on parallelism and the multifaceted nature of adaptation to a novel host environment.

### 4.1. Genetics of Performance Traits

Heritabilities for adult weight and development of *C. maculatus* reared on lentil were modest to low in the BC mapping populations (i.e., PVE = 0.08 to 0.38). Similarly low heritabilities for host-specific performance traits have been documented in other plant-insect systems [85], but larger heritabilities were expected here given the degree of adaptive divergence between M and the L lines [38,71]. Development time, and to a lesser extent adult weight, in each BC mapping population was considerably more similar to that of lentil-adapted lines than M line beetles reared on lentil [71]. This result is not entirely unexpected as each backcross was to a lentil line, but the degree of phenotypic similarity between the lentil and backcross lines exceeds additive expectations and suggests dominance of adaptive L-line alleles. Past work with line crosses suggested a mostly additive genetic architecture of survival in lentil with some dominance effects toward either the M (in L1 crosses) or L lines (in L2 crosses), but low survival of M-line beetles in lentil precluded estimates of dominance effects [106]. Moreover, reversals of dominance between sexes have been detected in *C. maculatus* for alleles associated with adaptation and life history [63]. Strong dominance effects for performance traits have been documented in other systems and are consistent with scenarios where a threshold level of enzymatic activity is necessary for detoxification or metabolism of a host plant (e.g., [107,108]). Future work with trait mapping in more variable mapping populations (e.g., M × L hybrid swarms) could better resolve the contributions of additive versus dominance effects to these performance traits.

Our results suggest that adult weight and development time had a polygenic basis in the BC mapping populations, with a lack of major effect loci, except perhaps for weight in BC-L2. A polygenic architecture is not unexpected given the quantitative and complex nature of weight and development time, and similar results have been observed in other systems (e.g., [13,85]). We found negative correlations between effect estimates for adult weight and development time, such that SNPs associated with increased weight were also associated with slower development (consistent with [62]). Correlations were especially large in BC-L1 and BC-L14 (i.e., |r|> 0.5). These high genetic correlations suggest many causal variants with pleiotropic effects on both performance traits or tight linkage and high LD between variants affecting the two traits. Distinguishing between these two possibilities is difficult, and from a functional and analytical perspective, true pleiotropy is best viewed as an endpoint on a continuum from loose to tight linkage [13].

In contrast, correlations in model-averaged effects for either trait between mapping populations were low. Genetic correlations based on genomic-estimated breeding values were sometimes higher, e.g., the genetic correlation between breeding values in BC-L14 based on the BC-L1 versus BC-L14 genotype-phenotype map was 0.17 for adult weight, but they were also idiosyncratic, with negative correlations occurring as often as positive ones. These results suggest that mostly different causal variants were segregating in the three mapping populations, or that causal variants were mostly in LD with different subsets of the sequenced SNPs or with different alleles at the SNP loci (negative genetic correlations indicate the latter was likely true in at least some cases). Another possibility is that epistatic interactions between genetic variants affecting weight or development time are prevalent, such that different combinations of alleles are favored at some of the same loci by selection on lentil (as in, e.g., [109,110,111]). Additional, larger crosses using populations with variable genetic backgrounds will be needed to evaluate these alternative (but not mutually exclusive) hypotheses. We discuss the implications of these findings for parallel evolution below.

### 4.2. Parallelism in Change Versus Traits

Population genomic analyses revealed considerable parallelism in patterns of genome-wide allele frequency change, and to a lesser extent in the specific SNPs exhibiting the highest rates of standardized allele frequency change, with notably higher parallelism between L1 and L14A than between either of these and L2. This is consistent with previous analyses of these experiments focused on estimating selection and testing for parallelism in selection [38,40], and with other studies where adaptation occurs from standing genetic variation (e.g., [21,27,29,112,113]). We observed considerably lower parallelism in allele frequency change in the reversion lines and in the genotype-phenotype associations for weight and development time. Lower parallelism in the reversion lines could be explained by weaker selection on mung bean or by greater differences in standing genetic variation, as each reversion line was derived from a distinct lentil line [114]. Notably, L1R and L2R show a lack of parallelism in loss of performance on lentil, with much lower survival rates in L1R than L2R [40,79].

Lower parallelism in the genetic architecture of the performance traits than in patterns of evolutionary change would not generally be expected because selection should only cause repeated patterns of change if the same genes or alleles affect traits under selection (whereas parallelism or repeatability is sometimes used as a test of selection, especially in E&R experiments, e.g., [33], selection does not necessarily result in parallelism and other means exist to detect selection, e.g., [40,115]). We think there are several, complementary explanations for this seemingly paradoxical result. First, other performance traits, most notably survival, evolve during rapid adaptation and could be the cause of parallel genetic changes. Support for this hypothesis comes from the limited overlap between SNPs associated with the measured performance traits and exceptional change genomic regions from the E&R experiments. Indeed, these sets of SNPs only overlapped more than expected by chance in the L1 line. Unfortunately, beetles that do not survive often die early, as small 1st instar larvae, and retrieving these beetles for genomic work is not practical, though alternative experimental designs are possible. Second, we likely captured (via LD) only a subset of the genetic variants affecting performance and weight in each mapping population, and missing causal variants (i.e., those not in LD with our SNPs) could be shared across lines. GWA studies commonly fail to detect causal variants, especially when sample sizes are not extremely large [116,117], and this limitation could be compounded here by possible dominance of lentil alleles (see above). Lastly, if genes harbor multiple functional variants in the source M population, the specific variants (and their effects) that are retained through the severe bottleneck that precedes lentil adaptation could vary across lines, and yet many of the same genes (or regions of the genome) could still be affected in a repeatable manner by selection. In other words, trait genetic architectures could be more sensitive to initial conditions set by the bottleneck (i.e., they could be more chaotic) than subsequent patterns of evolutionary change.

### 4.3. Genomics of Host Use and Adaptation

We found more evidence of evolved differences in gene expression among the M, L1 and L1R lines than of plastic differences caused by the host environment. Nonetheless, we detected consistent differences in expression of cytochrome P450s between mung bean and lentil treatments. Cytochrome P450s are known to play a role in detoxification of plant secondary metabolites and insecticides [53,101,118,119], and this finding is consistent with a general pattern of increased plasticity of detoxification genes in herbivorous insects [76,120]. Because mortality is high for the M line when reared in lentil, we do not know if adaptation to lentil involved the evolution of increased plasticity of cytochrome P450s or if this differential expression was present in the ancestral M line and thus is mostly incidental to lentil adaptation. However, some role of cytochrome P450s in lentil-adaptation is likely, as cytochrome P450 4d2 is among the genes showing evidence of parallel exceptional change across multiple lentil lines (L1 and L14A). There was some idiosyncratic and weak evidence of evolved differences in expression of digestive enzymes, which is again consistent with an emerging trend in analyses of herbivorous insects [76]. Moreover, *C. maculatus* has been shown to increase expression of proteases, including inhibitor-insensitive proteases, in response to plant protease inhibitors [75]. However, the gene with the most compelling signal was beta-glucosidase, which is involved in metabolism but also in detoxification because of its role in converting cyanogenic glycosides to toxic compounds [121]. Interestingly, lower enzymatic activity of a beta-glucosidase gene was shown to be associated with adaptation of *C. maculatus* to fava bean (*Vicia faba*, which is in the same tribe, Fabeae, as lentil) [122]. Specifically, low enzymatic activity of beta-glucosidase reduced conversion of vicine (a phytochemical produced by fava bean) to toxic aglycone. Exceptional change occurred in parallel (especially for L1 versus L14) in numerous other genes, some but not all of which are associated with metabolism or detoxification in *C. maculatus* (e.g., 5-oxoprolinase [103]).

In most cases, differentially expressed genes did not overlap more than expected with genetic loci associated with the performance traits or with regions of exceptional change during lentil adaptation (or reversion). The most notable exception was that genetic regions that showed exceptional changes in allele frequency during lentil adaptation in L1 overlapped more than expected by chance with genes showing plastic differential expression in L1 when reared in mung bean versus lentil. This is consistent with a role for the evolution of plasticity in host adaptation (as in, e.g., [54,120]). However, the lack of (excess) overlap among these three genomic datasets (allele frequency change, genotype-phenotype associations, and differential gene expression) is perhaps even more notable. This result suggests that the genetic basis of host adaptation, which is arguably most completely measured by genetic changes during adaptation, is not equivalent to the genetic architecture of performance traits or differential gene expression. Instead, each of these datasets provides a distinct and incomplete window into the genetics of adaptation. A lack of concordance among the different approaches is sobering. It may not be surprising because adaptation to lentil likely involves selection on numerous traits (or trait combinations), with each affected by multiple (and sometimes overlapping) genes, as well as by random changes in genes and traits during the extreme initial bottleneck on such a marginal host. Future experiments and genomic analyses in this system will build on the results presented here, with the eventual aim of producing a predictive genotype-phenotype-fitness map for *C. maculatus* adapting to lentil. Comparisons with other systems, especially those where host adaptation does not involve high mortality and a severe bottleneck or where host shifts also include interactions with competitors, mutualists, or predators (e.g., [123,124,125]), could help determine what aspects of the genotype-phenotype-fitness map are general versus specific to this system.

## Figures and Tables

**Figure 1 genes-11-00400-f001:**
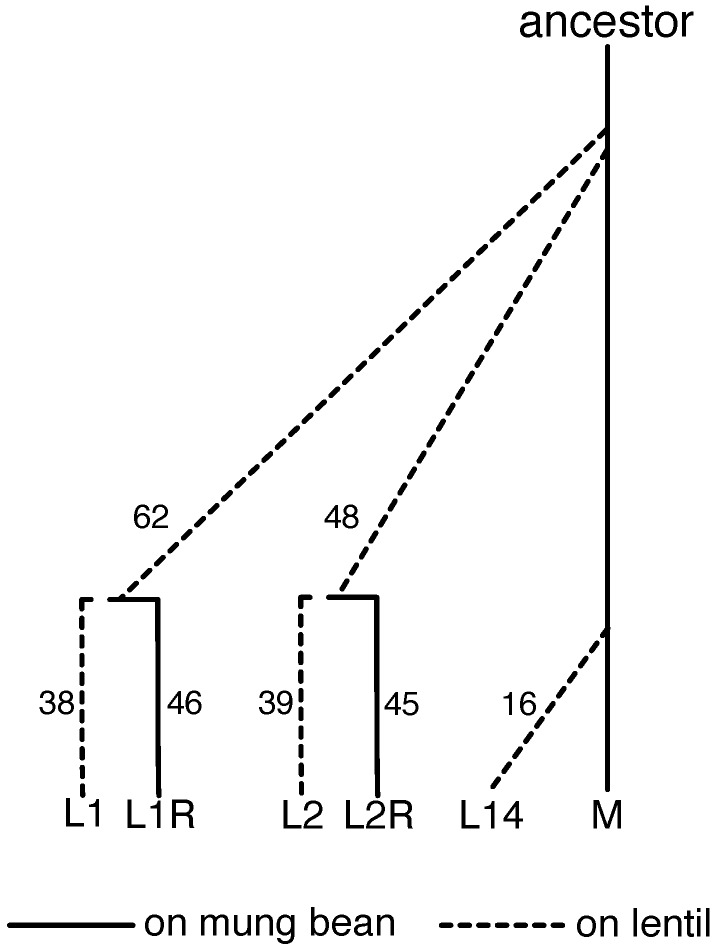
Illustration of the history of the *C. maculatus* lines discussed in this manuscript (i.e., L1, L2, L1R, L2R and L14) along with the South Indian mung line (denoted M). The number of generations that elapsed between the origin of each line and our final sample for population genomic analyses is shown. Details on additional samples including those used for the backcross mapping families and gene expression experiments can be found in Appendix A.

**Figure 2 genes-11-00400-f002:**
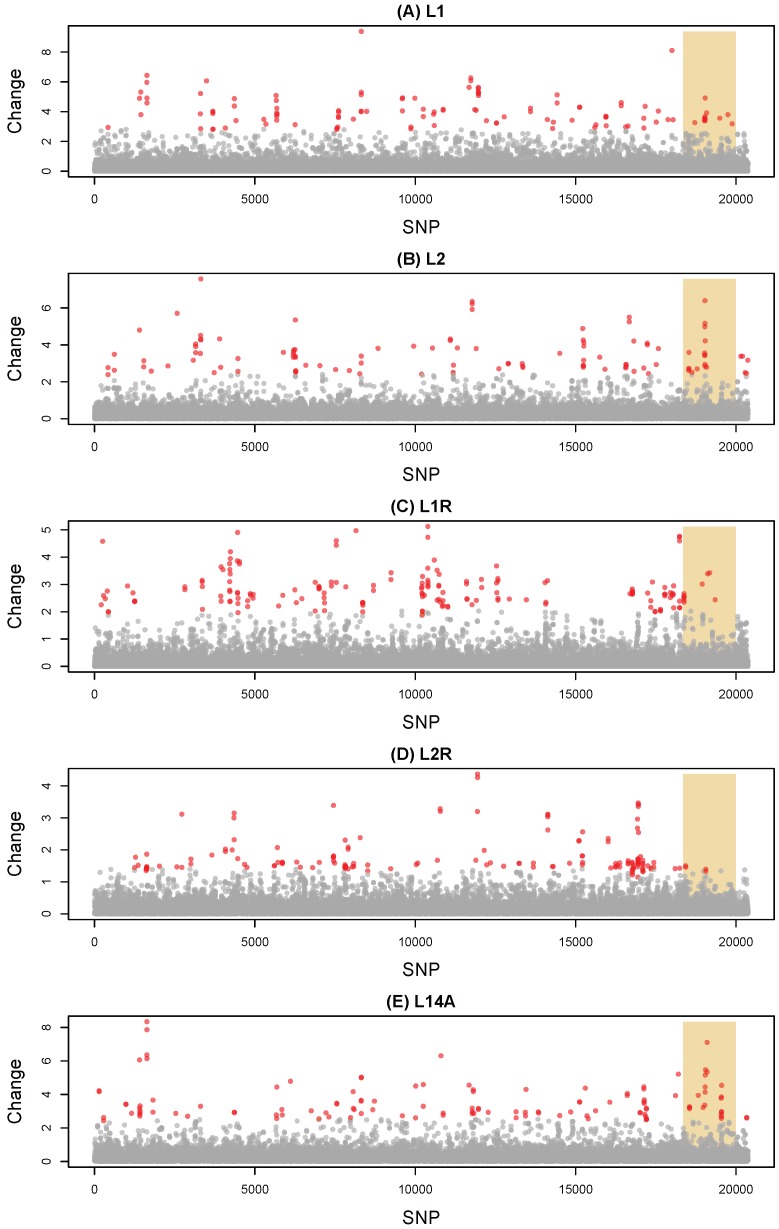
Manhattan plots show standardized allele frequency change for each of 20,376 SNPs for five pairs of lines and samples (that is, allele frequency change relative to the ancestral, expected heterozygosity). Results are shown for (**A**) M to L1 F100, (**B**) M to L2 F87, (**C**) L1 F91 to L1R F46, (**D**) L2 F78 to L2R F45, and (**E**) L14 P to L14A F16. Points denote change for individual SNPs, which are organized along the x-axis by scaffold and position within scaffold. The shaded region denotes putative X-linked SNPs, whereas SNPs to the left or right of this region are autosomal or Y-linked. Points are colored to reflect their hidden state from the HMM, with gray for average change and red for exceptional change.

**Figure 3 genes-11-00400-f003:**
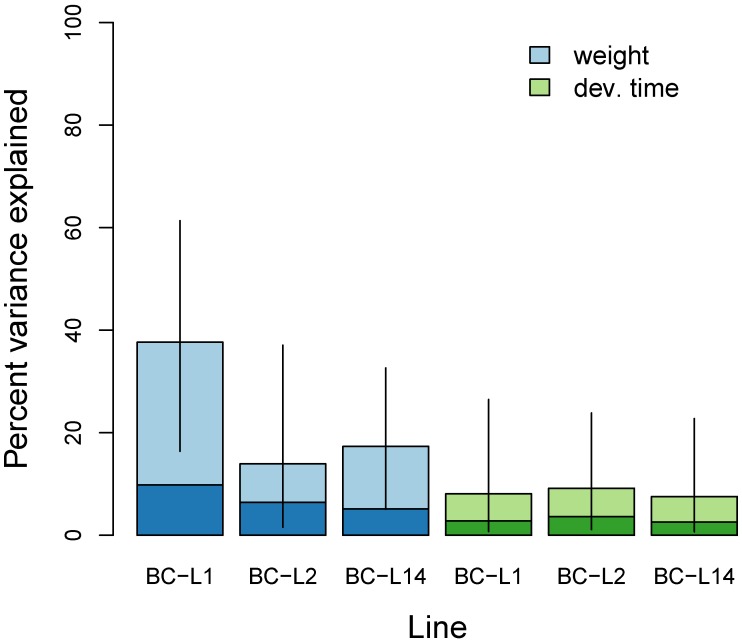
Bayesian estimates of the percent of variance in weight and development time explained (PVE) by the additive effects of genetic variants as captured by the SNP dataset for each of the three back-cross lines (BC-L1, BC-L2, and BC-L14). Colored bars denote point estimates of the PVE, and vertical lines denote 95% equal-tail probability intervals. The darker portion of each bar indicates a point estimate of the portion of the PVE attributable to genetic variants with individually measurable effects.

**Figure 4 genes-11-00400-f004:**
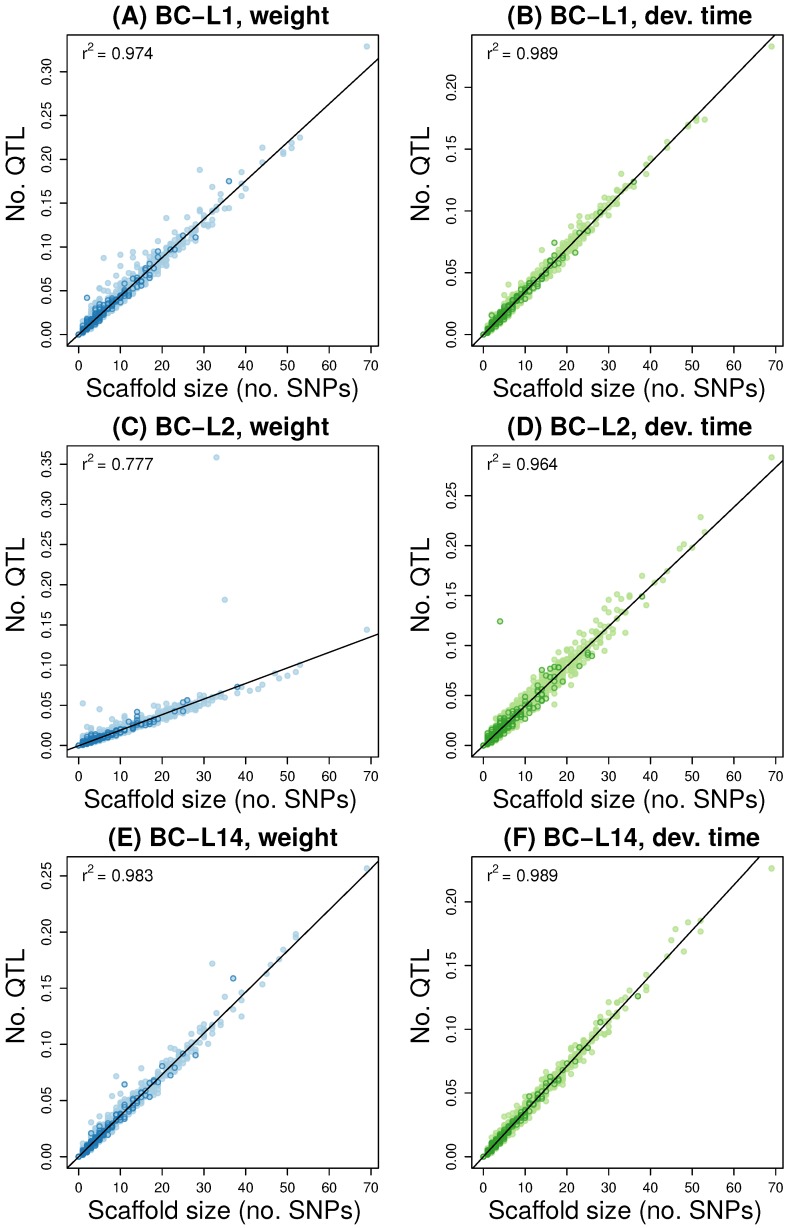
The scatter plots depict Bayesian estimates for the number of QTL (i.e., causal variants) on each scaffold as a function of scaffold size (here measured as the number of SNPs). Results are shown for each back-cross line (BC-L1, BC-L2 and BC-L14) and trait (weight [A, C, and E] and development time [B, D, and F]). Points representing X-linked scaffolds are denoted with dark outlines. A best fit line from linear regression is given and the coefficient of determination ($r^{2}$) is reported. Points that fall well-above the best fit line harbor more QTL than expected given their size and a highly polygenic architecture of very small effect variants scattered evenly across the genome.

**Figure 5 genes-11-00400-f005:**
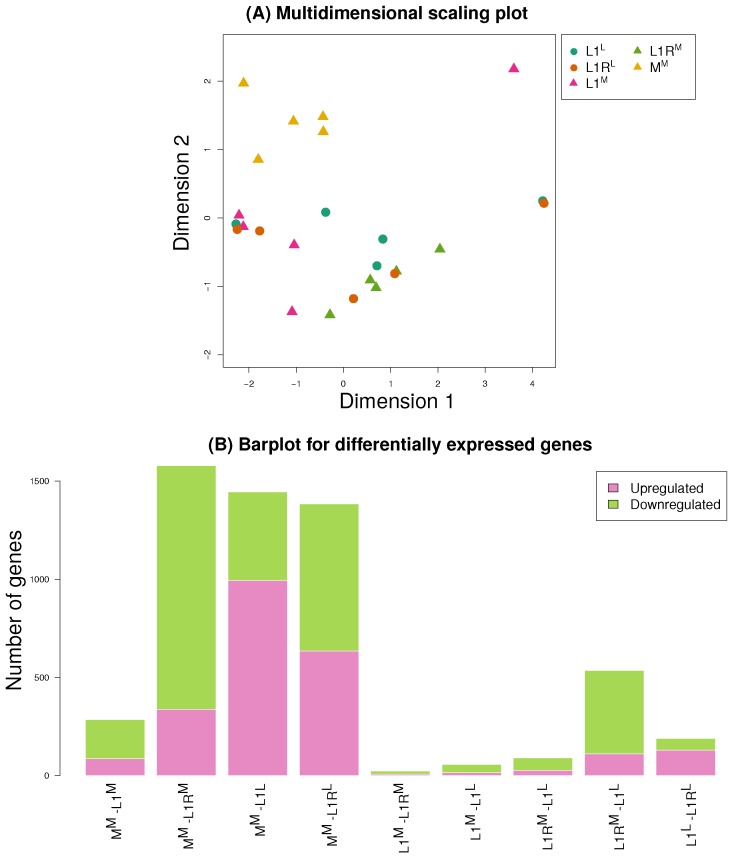
Summary of gene expression results. Panel (**A**) shows a NMDS ordination of the samples based on transformed count data. Points are labeled by line and host treatment. Bar plots in panel (**B**) depict the number of differentially expressed genes for each comparison, with colored regions denoting the number of upregulated versus downregulated genes in each contrast. Here, upregulated means that expression was higher in the first sample listed than in the second, and vice a versa for downregulated.

**Figure 6 genes-11-00400-f006:**
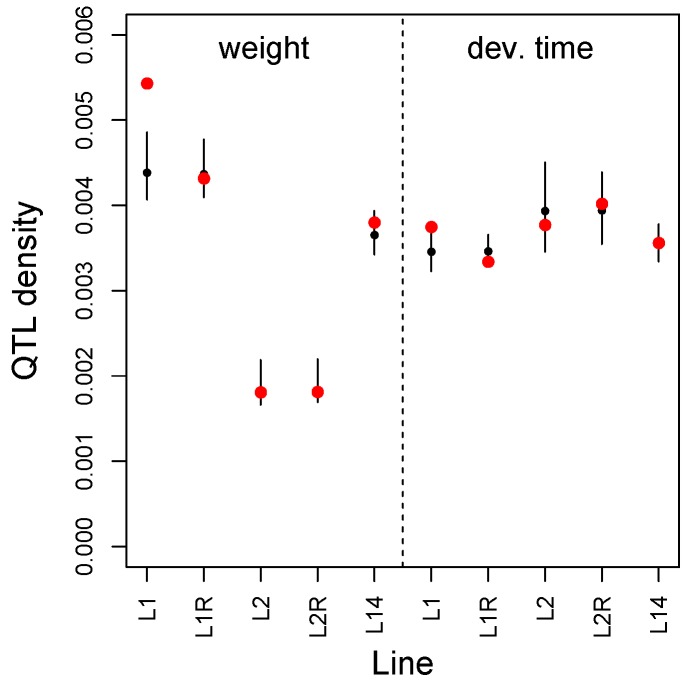
Performance-trait QTL density (number of QTL per SNP) for SNPs exhibiting exceptional allele frequency change in the E&R experiments. Results are shown for adult weight and development time for each pair of lines or samples (see main text for details). Red dots denote the observed density. Black dots (median) and vertical lines (2.5th to 97.5th percentile) denote the null expectations from randomization tests.

**Table 1 genes-11-00400-t001:** Summary of differential expression of cytochrome P450 enzymes. For each comparison, we report the number of differentially expressed cytochrome P450s, the expected number of differentially expressed cytochrome P450s given the number of differentially expressed genes in all and the proportion of genes classified as cytochrome P450s, and the binomial probability of having the observed number of differentially expressed cytochrome P450s by chance. The first three comparisons correspond to genetic differences in expression, the next two to plastic (host) differences in expression, and the final four include genetic and plastic differences.

Comparison	Observed	Expected	*P*
M${}^{M}$ × L1${}^{M}$	0	0.85	0.42
M${}^{M}$ × L1R${}^{M}$	3	4.93	0.14
L1${}^{L}$ × L1R${}^{L}$	0	0.57	0.57
L1${}^{M}$ × L1${}^{L}$	2	0.17	0.01
L1R${}^{M}$ × L1R${}^{L}$	6	1.67	0.01
M${}^{M}$ × L1${}^{L}$	7	4.51	0.08
M${}^{M}$ × L1R${}^{L}$	6	2.24	0.12
L1${}^{M}$ × L1R${}^{M}$	0	0.07	0.93
L1${}^{L}$ × L1R${}^{M}$	1	0.28	0.21

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
