# Peer review of "Combining Experimental Evolution and Genomics to Understand How Seed Beetles Adapt to a Marginal Host Plant"

_genes, 2020, doi:10.3390/genes11040400_

Round 1

Reviewer 1 Report

The manuscript titled “Combined experimental evolution and genomics to understand how seed beetles adapt to a marginal host plant” combines genomic analyses of evolve and resequence experiments, genotype-trait mapping of weight and development time, and differential gene expression to investigate the genetic architecture of host (lentil) adaptation in a pest of stored legumes and an emerging model system in experimental evolution, Callosobruchus maculatus. Contrary to the authors’ expectations very little overlap was observed in the SNPs associated with host adaptation, weight and development time, and differential gene expression patterns in comparisons of mung bean and lentil adapted lines. Despite this, the author’s present a thorough discussion of possible mechanisms/processes on which these results may be contingent, and identify genes with likely functional roles in host adaptation in this species. The manuscript is very well written, and I only found a few grammatical and/or spelling errors. These are listed below with other minor criticisms. One concern pertains to the SNP filtering steps used and the lack of explanation and/or rationale for these, especially given the leniency of the filters. The authors are encouraged to provide a rationale and support for the current filters ideally through parameterization experiments showing their relevance, but omission of these should not preclude publication in Genes. I recommend this manuscript be accepted with the minor revisions noted below.

Minor criticisms:

Lines 65-68 – The background information / rationale for including gene-expression data is lacking in comparison to that provided for the evolve and resequence and trait mapping aspects of the study. This sentence also appears to be somewhat tautological (eg. ‘a more comprehensive analysis would include gene-expession data for a more complete connection…’).

Line 151 – “had” not “has”.

Sections 2.2 and 2.3 – Provide references to the data archives for these data.

Line 233 – Provide the custom perl scripts either as supplementary material or as a link to a public forum such as GitHub.

Lines 251-253 – Provide a rationale for the filtering measures used. It is concerning that the authors have used such lenient SNP filters but have not performed any parameterizing (or indicated as such) to determine whether their filters have biased the results or not (for example by swamping out signal from loci with actual linkages to adaptation/causal traits). In my own experience, coverage levels of 2X and minor allele frequencies of 0.005 are really low, and might result in erroneous variant loci in the dataset (see O’Neal 2018 https://doi.org/10.1111/mec.14792 for a discussion on common sources of error in GBS datasets and filtering recommendations to address these).

Line 251 – Remove “based”

Lines 382-389 -  Provide an explanation on how gene identities were determined.

Line 436 – “pair” not “pairs”.

Line 526-528 – see comment above for Lines 251-253.

Lines 632-633 – This sentence is awkward. Perhaps change “showing” to “showed”.

Figures and Tables

In general, the text of the figures is very small and difficult to see at actual size (eg. I cannot read the legend in Figure S6). Font size should be increased.

Reviewer 2 Report

This study investigates adaptation to a novel environment in the seed beetle, Calosobruchus maculatus, an emerging model organism with a recently published genome that makes studies like this possible. This organism presents an advantage over other model organisms in that its laboratory culturing environment closely mimics its natural environment (it has been a pest of leguminous crops essentially since the dawn of agriculture), and the novel environment(s) that one could subject these beetles to in the lab (e.g. alternative bean/pea/seed species) are some of the exact conditions that they have apparently adapted to in nature. This study advances our understanding of local adaptation and will serve to put this emerging model system on the radar for others in the field.

The manuscript is well-written and easy to follow. I believe that my General and Specific comments (below) are mostly minor, and could be addressed with some adjusting of the language or presentation of methods and results in a way that is more upfront about the quality/caveats of the data and design. That said, the authors are mostly appropriately cautious/modest with their conclusions and do a good job of tying together a somewhat messy story into a sensible and concise discussion.

General comments:

  1. I wonder whether the term “parallel evolution” (or similar) is really appropriate in many of the cases where it is used in this manuscript. Replicates of a given experimental evolution (EE) regime are typically seen as crucially necessary to rule out drift in evolve and resequence experiments. Only those allele frequency changes that are consistent among *all* replicate EE lines can even be considered as potential, putative, candidate sites of interest. The simulations from Kofler et al. (2014) are very sobering in this respect (citation below). What you’re calling parallelism in some cases is the minimum threshold to meet for a set of loci to be suitable for some downstream analyses (e.g. your expression and GWA mapping parts). So, it seems strange to me to hear consistent changes among 2 EE lines (i.e. the absolute minimum amount of replication) referred to as parallel evolution. In my eyes, those are just the subset of changes, out of the total list, that are minimally reliable, and offer a jumping off point for the downstream analyses. The exceptional-change SNPs don’t need to be confirmed SNP-for-SNP across EE lines, of course, but maybe contig-for-contig (or something) since your analysis aims to weed out redundant linked hits and you end up with an average ~1 SNP/contig for those exceptional-change SNPs. So, for example, maybe contigs harboring an exceptional-change SNP in BOTH the M to L1 F100 and M to L2 F87 could be considered candidate contigs that likely harbor loci that are involved in lentil adaptation. I could be wrong or confused about all of this, but I’m afraid your statistical approaches don’t fully compensate for the low (or lack of) replication of the experiment, on their own. If I’m wrong or confused about, then there should be some more effort to emphasize why your study design and/or analysis enables you to rule out drift to a greater degree, or in a different way, than similar studies that use many more replicates EE lines.
  2. That the severe bottle-neck of the populations in their initial phase of adapting to lentils is likely to prevent finding “parallel evolution” (or what I’m calling replication of findings) is partly addressed at the end of the Discussion, however, what is not addressed is that this bottle-neck could be the soul explanation for any allele frequency differences in the before/after adaptation comparisons for L1, L2, or L14A, which would in turn explain why many of the findings are not replicable/parallel across lines. The ideal control for those adaptation lines is not M, but actually replicated M populations that were randomly bottle-necked to a similarly severe degree as the adaptation lines. In fact, is it not an explicit assumption of the models you’re using to identify exceptional-change SNPs that the population sizes have remained constant throughout? These issues, and/or their counter-arguments, should be featured in the discussion and methods where appropriate.
  3. I think the phrase “exceptional change SNPs” should be hyphenated like: “exceptional-change SNPs”, or some alternative.
  4. It would be nice to know the approximate population size of the South India population throughout its laboratory maintenance after being collected from nature, so that readers know how bottlenecked it was or wasn’t from the start. This is important when it comes to estimating genetic variance.

Specific comments:

Line 52: strike the word “adequate” to make this a more accurate statement from a pop gen statistical power perspective. The level of replication needed to rule out random genetic drift is sobering, but unachievable. So we just have accept that our achievable levels of replication are the best we can do with some organisms, and then fix our statements and conclusion accordingly.

-Kofler, R. and Schlötterer, C., 2014. A guide for the design of evolve and resequencing studies. Molecular biology and evolution31(2), pp.474-483.

Line 78: These are good papers, but I think it would be better for the sexual conflict citations to pertain to the *genetic* conflict between the sexes (i.e. intra-locus sexual conflict, a.k.a., sexually antagonistic genetic variation), since your paper is more to do with identifying the causal genetic variation underlying adaptation. The following 3 citations are more relevant to the “sexual conflict” that I would think you’re referring to here, as well as to other aspects of your study/discussion (use them if you find them useful).

This study shows how that sexually antagonistic genetic variation is selected on during adaptation to novel environments.

-Berger, D., Grieshop, K., Lind, M.I., Goenaga, J., Maklakov, A.A. and Arnqvist, G., 2014. Intralocus sexual conflict and environmental stress. Evolution68(8), pp.2184-2196.

This study shows the suite of life history traits that are affected by this sexually antagonistic genetic variation, where mass and development time scale oppositely for their life-history principal component (akin to your negative correlation for those traits (Line 554-556)).

-Berger, D., Martinossi-Allibert, I., Grieshop, K., Lind, M.I., Maklakov, A.A. and Arnqvist, G., 2016. Intralocus sexual conflict and the tragedy of the commons in seed beetles. The American Naturalist188(4), pp.E98-E112.

Lastly, this study demonstrates that those sexually antagonistic alleles – involved in adaptation and life history – exhibit a polygenic signal of reversed dominance between the sexes, which may help your Discussion of additive versus dominance effects in performance traits (Line 540-550).

-Grieshop, K. and Arnqvist, G., 2018. Sex-specific dominance reversal of genetic variation for fitness. PLoS biology16(12), p.e2006810.

Line 137: What is the “A” in “L14A”?

Line 153: The question I asked for Line 137 is answered on this line. This would ideally be clarified earlier, or the “A” removed in Line 137.

Line 156: I have no idea what “fitness” actually means here. That the ne can be maintained at >2000?

Line 162: This is probably more of an issue on my end than on yours, but there’s not really enough here for me to understand what kind of data you have. I think that for each generation for which you have sequencing data, you have a pool of 992 beetles from 22 “lines”, which, depending on what “line” means, is probably a lot closer to 44 genomes (since presumably it took a male and female to found each line) in your pooled sample than it is to 992 genomes. I then find it even less understandable, on Line 251, that you filter the data by ~2X coverage *per beetle*. I’m basically trying to figure out how much coverage you have in your data, after filtering, so that I can have a rough feel for your accuracy to call minor alleles of frequency ~0.005 (Line 253), and moreover your power to detect allele frequency changes of a given magnitude. Perhaps you should just state these things straightforwardly so that readers don’t have to figue this out on their own… I am thinking of this resource:

Lynch, M., Bost, D., Wilson, S., Maruki, T. and Harrison, S., 2014. Population-genetic inference from pooled-sequencing data. Genome biology and evolution6(5), pp.1210-1218.

Line 167-182: This backcrossing scheme is a little hard to follow. Line 168 reads as if you put 2 females (n *pairs* of females) with each male, and then it sounds like you’re moving onto the next stage, but then Line 172 starts “Individual pairs…”. Are you returning to the description of this first step at the start of a new paragraph after briefly describing how you obtained these unmated individuals? Or are these two different steps? Can’t quite tell since the first time it’s 2 females/male and the next time its “Individual pairs…”

Line 188-189, and section 2.6 (GWA mapping methods): So the development time phenotype has both males and females in it? Without ‘sex’ as a fixed effect, I guess the development data could be bimodally distributed, since male and female C. maculatus have different development times. If the data are bimodal, it would be a problem for estimating the variance in development time via REML (Line 307). It would probably also affect your Bayesian approach (Line 311) to identifying associated SNPs and their model inclusion probabilities. I realize that the Bayesian approach should accommodate non-ideal data, but it sounds like the SNP effects are being estimated as a latent variable (I might be wrong about that), and so the model’s ability to associate SNPs with development time is only as good as the estimation of those family-level highest posterior density mean (or mode) development times. Is there a way to assess this? Are the credibility intervals for the family-level posteriors useful (i.e. are the HPD means estimated well(?), or are their CIs really wide?). The best approach, for both the REML and Bayesian models, would be to modeled this otherwise unexplained noise by fitting the fixed effect of ‘sex’. Alternatively (if that option isn’t available), you could divide each observation’s development time by the mean development time within its respective sex, to remove the sex effect from the raw data prior to modeling or GAW mapping.

Line 190-201: So was there a library prepared for each family? The S.D. on those mean number of beetles per back-cross line are really high (relative to the mean), so I guess that means that some libraries for some families are based on only a couple or a few beetles, correct? Is that an issue? I’m thinking small numbers in some families could inflate the between-family genetic variance, which would in turn affect the GWA mapping, right?

Line 265-276: Why not have “L14 P to L14A F16 (change during lentil adaptation for L14)” be contrast number iii, so that your predictions on 274-276 can be with respect to contrasts i-iii (adaptation to lentil), and iv-v (reversion to mung)?

Line 412: Close parentheses after the second “mung”.

Fig. 4: It might make the figures a little more useful to somehow identify those scaffolds that harbor more QTL than expected from the number of SNPs/scaffold… Maybe by coloring them or outlining them differently or something.

Fig. 4 caption (and where ever else it applies): Regarding the phrase “…and a highly polygenic architecture…”, are you confident that a cluster of QTL on a given scaffold represents many independent QTL and not just a single true QTL and some other linked signals in the vicinity of that scaffold? There’s not a recombination map, right? I guess my imagination of a “highly polygenic architecture” would look like many different scaffolds being modestly above this best-fit line, not a few of them being far above the line.
